# Non-Invasive Position Measurement of a Spatial Pendulum Using Infrared Distance Sensors

**DOI:** 10.3390/s25154624

**Published:** 2025-07-25

**Authors:** Marco Carpio, Julio Zambrano, Roque Saltaren, Juan Cely, David Carpio

**Affiliations:** 1Grupo de Investigación en Interacción Robótica y Automática (GIIRA), Universidad Politécnica Salesiana, Calle Turuhuayco 3-69 y Calle Vieja, Cuenca 010105, Ecuador; jzambranoa@ups.edu.ec; 2Grupo de Investigación en Energias (GIE), Universidad Politécnica Salesiana, Calle Turuhuayco 3-69 y Calle Vieja, Cuenca 010105, Ecuador; dcarpioj1@est.ups.edu.ec; 3Centro de Automática y Robótica, Universidad Politécnica de Madrid, C/José Gutiérrez Abascal 2, 28006 Madrid, Spain; roquejacinto.saltaren@upm.es; 4Intelligent Robotics Lab, Rey Juan Carlos University Fuenlabrada, 28942 Fuenlabrada, Spain; juan.cely@urjc.es

**Keywords:** spatial pendulum, non-invasive sensing, pendulum position measurement, physical sensor

## Abstract

For the study and experimentation of physical systems, it is essential to measure the physical variables, which implies choosing the most convenient method that does not affect the natural behavior of the system. This work presents the modeling and sensing of the spherical pendulum, integrating a novel non-invasive measurement scheme based on infrared sensors arranged in a quadrature configuration. The proposed method enables the estimation of angles around two axes, leveraging light reflection on a perpendicular plane aligned with the pendulum bar. A mathematical model was developed to create simulations, and a prototype was constructed to perform experiments and validate the detection method. The values recorded by the sensors enable the reproduction of the pendulum’s trajectory, allowing for the correlation of real results with those of the simulations. The similarity of behavior between the simulations and the experimentation facilitates the validation of the proposal.

## 1. Introduction

Precise measurement of physical variables is a fundamental aspect in the study, modeling, and control of dynamic systems. Depending on the nature of the physical system under analysis, it is necessary to select a sensing method that does not alter its natural behavior. This is particularly important in nonlinear and oscillatory systems such as the spherical pendulum, where the interaction between forces and degrees of freedom can be complex [1,2]. Pendulum-based systems have been widely used as experimental platforms for validating control strategies and modeling the dynamic behavior of mechanical systems in both terrestrial and space environments [3,4]. However, conventional angle measurement techniques are often invasive or may influence system behavior due to mechanical contact, wiring, or the added mass of sensors [5,6]. Specifically, methods incorporating potentiometers and encoders have been widely used due to their simplicity and high accuracy in control and monitoring applications [7,8]. These devices present significant limitations in terms of invasiveness and long-term stability. In particular, potentiometers require direct physical contact, which makes them invasive solutions, especially problematic in biomedical or high-sensitivity applications, where contact may alter the system or compromise its integrity [9,10]. Additionally, both potentiometers and encoders are susceptible to mechanical wear, contaminant accumulation, and temperature fluctuations, which can impact accuracy and necessitate periodic maintenance or calibration to maintain optimal performance [8].

To address these limitations, alternative angular measurement approaches based on optical technologies and non-invasive sensors have emerged, demonstrating high precision in environments where minimal system disturbance is critical [11,12,13]. In particular, light-reflection-based methods have gained interest due to their ability to measure angular displacements without physical contact, which is advantageous for high-sensitivity systems such as space inertial sensors or experimental setups involving complex dynamic models [14,15]. These sensors, especially those based on infrared spectroscopy or proximity beams, have demonstrated high accuracy comparable to that of traditional methods, with the advantage of eliminating the physical wear and tear inherent to contact systems. However, their long-term stability can be affected by instrumental drift and environmental conditions, so the implementation of online calibration and dynamic compensation techniques, such as orthogonal projection, becomes essential to maintain their reliability in special applications [10,16].

This work presents a novel non-invasive measurement technique for a spatial pendulum, based on an orthogonal arrangement of infrared sensors along the X and Y axes. These sensors detect the distance to a circular plane attached perpendicularly to the pendulum rod. This configuration enables the capture of the three-dimensional motion of the pendulum tip via light reflection without interfering with its natural oscillation [17,18]. Based on these measurements, a forward and inverse kinematic procedure is employed using the Denavit–Hartenberg method [19,20], which is widely used in the modeling of robotic manipulators and multibody systems [21,22]. This allows real-time inference of the pendulum’s angular orientation relative to the principal axes.

The dynamic model of the system was developed using a Lagrangian approach, incorporating a damping term that aggregates real-world effects such as friction at the pivot, medium density, and geometric dissipation [23,24]. This term was experimentally calibrated by analyzing the amplitude decay during prolonged oscillation tests and subsequently incorporated into the simulation framework [18].

The experimental implementation was validated through a physical prototype, considering two motion scenarios: a planar oscillation and a circular-elliptical motion induced by specific initial conditions. In both cases, the comparison between simulation results and experimental recordings validated the proposed model, confirming the effectiveness of the sensing method and the relevance of the mathematical modeling [25,26].

While the proposed measurement system is functional, possible improvements in measurement quality have been identified, such as digital filtering of the signals, sensor positioning accuracy, and the use of industrial-grade components for future implementation in aerospace or robotic applications [27,28,29].

## 2. Mathematical Modeling of the Spatial Pendulum

The system is modeled in terms of the generalized coordinates (θ, ϕ), where θ represents the inclination angle relative to the vertical axis, and ϕ denotes the azimuthal angle, as illustrated in Figure 1.

The length of the pendulum is denoted by *l* and the mass by *m*. The position equations of the mass in Cartesian coordinates are given by:(1)x=lsinθcosϕ,(2)y=lsinθsinϕ,(3)z=−lcosθ,

From the time derivative of the position vector, the magnitude of the velocity vector is obtained as:(4)$$v^{2}=l^{2}\left({\dot{\theta }}^{2}+{\dot{\phi }}^{2}sin^{2}\theta \right),$$

The kinetic and potential energies of the system are:Kinetic energy(5)$$T=\frac{1}{2}ml^{2}\left({\dot{\theta }}^{2}+{\dot{\phi }}^{2}sin^{2}\theta \right),$$

Potential energy


(6)
V=−mglcosθ,


The Lagrangian [23] is defined as:(7)$$L=\frac{1}{2}ml^{2}\left({\dot{\theta }}^{2}+{\dot{\phi }}^{2}sin^{2}\theta \right)+mglcos\theta ,$$

By applying the Lagrange equations [23], Equations (Equation 8) and (Equation 9) are obtained, which represent the dynamics of the spherical pendulum with conservative characteristics:(8)$$\ddot{\theta }-{\dot{\phi }}^{2}sin\theta cos\theta +\frac{g}{l}sin\theta =0,$$(9)$$\ddot{\phi }-\frac{2\dot{\theta }\dot{\phi }}{tan\theta }=0,$$

It is important to note that the pendulum, in practical and real scenarios, operates within an environment other than a vacuum (in this case, air); therefore, it is subjected to a non-conservative force that progressively reduces its energy. This force primarily combines the effect of air friction acting on the pendulum’s geometry, influencing the inertial effects that produce ideal oscillation. To adjust the ideal mathematical model proposed by Equations (Equation 8) and (Equation 9), a new term dependent directly on the angular velocities and a damping coefficient β, representing the frictional effects of the medium, is introduced. Equations (Equation 10) and (Equation 11) represent the real-world pendulum dynamics:(10)$$\ddot{\theta }={\dot{\phi }}^{2}sin\theta cos\theta -\frac{g}{l}sin\theta -\beta \dot{\theta },$$(11)$$\ddot{\phi }=-\frac{2\dot{\theta }\dot{\phi }}{tan\theta }-\beta \dot{\phi },$$

To simulate the dynamic behavior of the pendulum, a state-space system is constructed from Equations (Equation 10) and (Equation 11), consisting of four first-order nonlinear differential equations represented by Equations (Equation 12) through (Equation 15). The state variables [θ, ϕ, ω, ψ] correspond to the angular positions and angular velocities for each axis of motion.(12)$$\dot{\theta }=\omega ,$$(13)$$\dot{\phi }=\psi ,$$(14)$$\dot{\omega }=\psi^{2}sin\theta cos\theta -\frac{g}{l}sin\theta -\beta \omega ,$$(15)$$\ddot{\phi }=-\frac{2\omega \psi }{tan\theta }-\beta \psi ,$$

These equations encompass the pendulum’s dynamics, whose nonlinear characteristics can be calculated and simulated using mathematical software.

## 3. Simulations

The implementation of the differential equations of motion (Equation 12) to (Equation 15) is carried out in MATLAB software. By calculating the derived states [θ, ϕ, ω, ψ] through the numerical resolution method of Euler, the values of the states at each time interval are obtained consecutively, as described in Equations (Equation 16) to (Equation 19).(16)$$\theta_{i}={\dot{\theta }}_{i}\Delta +\theta_{i-1},$$(17)$$\phi_{i}={\dot{\phi }}_{i}\Delta +\phi_{i-1},$$(18)$$\omega_{i}={\dot{\omega }}_{i}\Delta +\omega_{i-1},$$(19)$$\psi_{i}={\dot{\psi }}_{i}\Delta +\psi_{i-1},$$

Let Δ be the sampling interval, where _*i*_ represents the current value of the variable and _*i*−1_ represents the previous value computed by the algorithm. For the simulation, the system parameters are considered as follows:g=9.8$\frac{m}{s^{2}}$, gravitational acceleration;l=0.3 m, pendulum length;m=0.1 kg, pendulum mass;β=0.01, damping coefficient;Δ=0.05 s, time step for numerical calculation.

The damping coefficient was estimated from experimental data and validated against simulation results to ensure equivalent dynamic behavior. A long-duration experiment was performed to measure the decay in oscillation amplitude from real measurements. Based on this, a damping factor β was sought such that when introduced into Equations (Equation 14) and (Equation 15), it yields similar results. This topic is addressed further in the experimental section.

Two cases are considered for simulation: Case 1 involves initial angular displacements in θ and ϕ, inducing oscillatory motion of the pendulum. Case 2 includes initial angular displacements in θ and ϕ plus an initial angular velocity $\dot{\phi }$. These conditions theoretically generate a conical oscillation, allowing the evaluation of the pendulum’s position component behaviors.

### 3.1. Case 1: Initial Angular Displacements Results

For an initial condition of [θ, ϕ, ω, ψ] = [10°, 45°, 0 rad/s, 0 rad/s], over 200 iterations, the components generating the pendular oscillatory motion in a plane defined by a 45-degree diagonal between the XY base plane and the Z-axis (as shown in Figure 1) were calculated. In this case, the amplitude of the pendulum displacement along each axis exhibits oscillatory behavior with a sinusoidal character. These effects are illustrated in Figure 2, respectively.

Additionally, to provide a better spatial understanding of the pendulum’s trajectory, Figure 3 is presented.

It is noteworthy that due to the decay in oscillations, the pendulum’s tip progressively loses height, which begins to be noticeable in the final samples of Figure 2c, given that the damping effect on the pendulum is not very significant.

Observing Figure 3a, it can be noted that the pendulum generates oscillation in a plane formed by a diagonal in the base XY plane and the *Z* axis. Regarding Figure 3b, which presents a top view of the base XY plane, it can be seen that due to the absence of an initial angular velocity $\dot{\phi }$, the motion follows the 45-degree diagonal defined by the initial condition of the angle ϕ.

### 3.2. Case 2: Initial Angular and Velocity Displacements Results

Assuming the initial conditions [θ, ϕ, ω, ψ] = [10°, 45°, 0 rad/s, 2 rad/s] and a total of 1000 iterations, it is noteworthy that introducing an initial angular velocity of $\dot{\phi }$ = 6 rad/s results in a helical concentric motion. This behavior arises as the displacement amplitudes along the *X*, *Y*, and *Z* axes vary according to sinusoidal functions and progressively attenuate due to damping effects. These dynamics are illustrated in Figure 4, respectively. As in the previous case, and to enhance spatial visualization of the pendulum’s trajectory, a three-dimensional representation is provided in Figure 5.

As observed in Figure 5a, which presents the oscillatory motion in the XZ plane, the pendulum exhibits a progressively decreasing oscillation due to frictional forces from the surrounding medium. This damping effect drives the system toward a steady-state condition, where the pendulum stabilizes in a vertical position associated with a lower height. In Figure 5b, it is evident that the introduction of an initial angular velocity $\dot{\phi }$ results in a concentric motion in the XY plane, which gradually attenuates over time as the number of samples increases. Finally, Figure 5c offers a spatial perspective of the pendulum’s behavior, where the oscillatory pattern reveals a conical motion with an elliptical tendency.

## 4. Non-Invasive Sensing Using Infrared Sensors

To accurately contrast the behavior of the pendulum, it is necessary to measure the angular positions of the pendulum directly or indirectly. For this reason, the method of sensing using infrared distance sensors will be employed, as described below.

### 4.1. Method of Sensing Justification

Conventional methods for measuring oscillation angles include potentiometers, optical encoders, and inertial sensors, each with inherent limitations, the most relevant being:Potentiometers introduce mechanical friction and wear.Optical encoders require direct contact with the pendulum shaft.Inertial sensors (IMUs) face challenges related to power supply or data transmission and are prone to cumulative errors or reference drifts during extended measurements [13].

The proposed quadrature infrared sensing approach enables accurate and contactless angle measurement, thereby avoiding these drawbacks.

### 4.2. Sensing System Configuration

It is mainly based on the following guidelines:Two infrared sensors are arranged in quadrature near the support point of the pendulum rod.Each sensor provides an analog output proportional to the intensity of light reflected from a flat surface mounted perpendicularly to the pendulum rod.

Figure 6 illustrates the pendulum system and the integration of the sensing mechanism.

Based on Figure 6, it can be observed that the placement of sensors Sy and Sx, aligned along the *Y*-axis and *X*-axis, respectively, enables the measurement of the distance to the reflective plane P by emitting and receiving infrared light along the negative *Z*-axis. The reflective plane P is rigidly attached to the pendulum rod and moves freely with its motion. The measured distance allows for the estimation of the angular rotation around the corresponding axis. To establish the mathematical relationship between the distance measured by the sensors and the angular displacement, an analysis of the projected components onto the reflective plane is conducted, as illustrated in Figure 7. This figure considers an absolute rotation $\theta_{1}$ about the fixed *X*-axis and a rotation $\theta_{2}$ about the fixed *Y*-axis.

Figure 8 details the dimensions of segments that define the reflection points. In this way, the orthogonal segments $l_{1}$ and $l_{2}$, together with the sensing plane distance $l_{p}$ from the pendulum’s rotation point, intersect the vector ${\mathrm{dz}}_{1}$, which corresponds to the distance measured by sensor Sx. Similarly, the segments $l_{3}$, $l_{4}$, and $l_{p}$ intersect ${\mathrm{dz}}_{2}$, which corresponds to the distance measured by sensor Sy.

#### 4.2.1. Generating the Distance Measured by Sensor Sx

The segments involved in generating the distance measured by sensor Sx are detailed in Figure 9. In this figure, the coordinate axes are assigned starting from the pendulum’s anchor point, defined as the fixed reference frame ($X_{0}$, $Y_{0}$, $Z_{0}$). Subsequently, a series of intermediate frames is implemented to register the successive rotations, culminating in the mobile reference frame ($X_{4}$, $Y_{4}$, $Z_{4}$).

From the analysis presented in Figure 9, the Denavit–Hartenberg (DH) parameters are derived and summarized in Table 1. This DH formulation includes the transformation between intermediate frames as follows:The rotation from frame {2} to frame {3} involves a transformation $R(X_{2},-\pi /2)$, which corresponds to a rotation of −π/2 radians about the $X_{2}$ axis.Similarly, the rotation from frame {3} to frame {4} involves a transformation $R(Z_{3},-\pi /2)$, indicating a rotation of −π/2 radians about the $Z_{3}$ axis.

These transformations are explicitly depicted in Figure 9, forming the basis for the kinematic model used to relate sensor measurements to angular displacements.

It should be noted that each row of parameters allows for establishing the homogeneous transformation matrix between intermediate coordinate frames in the form _*i*−1_$A^{i}$, as presented in Equation (Equation 20) and [19,20], where i is the index of the coordinate system being analyzed. Thus, the first row of Table 1 corresponds to the transformation matrix analysis between coordinate frame zero and coordinate frame one, with the analysis subsequently extended for each parameter row in the table.(20)Aii−1=r11r12r13pxr21r22r23pyr31r32r33pz0001,
where the coefficients *r* form the rotation matrix, while the vector [$p_{x}$,$p_{y}$,$p_{z}$] represents the position of the coordinate system {i} relative to the coordinate system {i−1}.

#### 4.2.2. Generating the Distance Measured by Sensor Sx

The segments that generate the distance measured by the sensor Sy are detailed in Figure 7. In this figure, the axes are assigned starting from the pendulum’s pivot point, with fixed axes ($X_{0}$, $Y_{0}$, $Z_{0}$). Subsequently, intermediate axes are implemented, registering the rotations performed until the mobile axes ($X_{4}$, $Y_{4}$, $Z_{4}$) are reached.

The Denavit-Hartenberg parameter table resulting from the analysis of Figure 10 is shown in Table 2. The Denavit-Hartenberg analysis includes the rotation of the intermediate coordinate system from {2} to {3} considering $R(X_{2}$,π), which implies a rotation of π radians about the $X_{2}$ axis. Additionally, the rotation of the intermediate coordinate system from {3} to {4} considers $R(Z_{3}$,–π/2), implying a rotation of –π/2 radians about the $Z_{3}$ axis, as specified in Figure 10.

By substituting each row of parameters from Table 1 into the Denavit–Hartenberg transformation matrix in Equation (Equation 21), the matrices relating the position and orientation of the consecutive axes in Figure 9 are obtained. Subsequently, the product of these matrices yields a total transformation matrix between system zero and system four, as shown in Equation (Equation 22). Similarly, by using the information from Table 2 and the mathematical expression (Equation 21), and following the same procedure described, the total transformation matrix corresponding to the axis configuration shown in Figure 10 is obtained, as presented in Equation (Equation 23).(21)$$A=\left[\begin{array}{cccc} cos\theta & -sin\theta cos\alpha & sin\theta sin\alpha & acos\theta \\ sin\theta & cos\theta cos\alpha & -cos\theta sin\alpha & asin\theta \\ 0 & sin\alpha & cos\alpha & d\\ 0 & 0 & 0 & 1 \end{array}\right],$$

Based on the formulation in Equation (Equation 20), the vector [$p_{x}$,$p_{y}$,$p_{z}$]’ from the matrix A40 represents the position of the moving axes relative to the fixed axes ($X_{0}$, $Y_{0}$, $Z_{0}$). It is essential to highlight that this vector is directly related to the position measured by each sensor. Therefore, for a given value of $l_{p}=0.03$ m, position vectors are obtained for each part of the Denavit–Hartenberg analysis previously performed, resulting in the mathematical expressions (Equation 22) and (Equation 23), respectively.(22)$$\left[\begin{array}{c} l_{p}\;cos\left(\theta_{2}+\frac{pi}{2}\right)+l_{2}\;sin\left(\theta_{2}+\frac{pi}{2}\right)\\ l_{2}\;cos\left(\theta_{1}+\frac{pi}{2}\right)cos\left(\theta_{2}+\frac{pi}{2}\right)-l_{1}\;sin\left(\theta_{1}+\frac{pi}{2}\right)-l_{p}\;cos\left(\theta_{1}+\frac{pi}{2}\right)sin\left(\theta_{2}+\frac{pi}{2}\right)\\ l_{1}\;cos\left(\theta_{1}+\frac{pi}{2}\right)-l_{p}\;sin\left(\theta_{1}+\frac{pi}{2}\right)sin\left(\theta_{2}+\frac{pi}{2}\right)+l_{2}\;cos\left(\theta_{2}+\frac{pi}{2}\right)sin\left(\theta_{1}+\frac{pi}{2}\right) \end{array}\right],$$(23)$$\left[\begin{array}{c} l_{p}\;cos\left(\theta_{2}+\frac{pi}{2}\right)+l_{4}\;sin\left(\theta_{2}+\frac{pi}{2}\right)\\ l_{3}\;sin\left(\theta_{1}+\frac{pi}{2}\right)-l_{p}\;cos\left(\theta_{1}+\frac{pi}{2}\right)sin\left(\theta_{2}+\frac{pi}{2}\right)+l_{4}\;cos\left(\theta_{1}+\frac{pi}{2}\right)cos\left(\theta_{2}+\frac{pi}{2}\right)\\ l_{4}\;cos\left(\theta_{2}+\frac{pi}{2}\right)sin\left(\theta_{1}+\frac{pi}{2}\right)-l_{3}\;cos\left(\theta_{1}+\frac{pi}{2}\right)-l_{p}\;sin\left(\theta_{1}+\frac{pi}{2}\right)sin\left(\theta_{2}+\frac{pi}{2}\right) \end{array}\right],$$

Equation (Equation 22) was obtained from the mathematical analysis of Figure 9, so each of its elements will be equivalent to each of the components of the vector [$dx_{1}$,0,−$dz_{1}$] corresponding to the position of a point of reflection of infrared light from the sensor Sx in the *P* plane, being its first element $dx_{1}$ representing the location of the sensor in the direction of the $X_{0}$ axis. Likewise, the second element corresponds to the component of the sensor position in the direction of the $Y_{0}$ axis, and the third element corresponds to the negative distance $dz_{1}$ in the direction of the $Z_{0}$ axis. The result of this mathematical procedure generates a system of three equations with six unknowns, which are the parameters [$\theta_{1}$,$\theta_{2}$,$l_{1}$,$l_{2}$,$l_{3}$,$l_{4}$].

Similarly, the procedure for the mathematical expression (Equation 23) referred to in Figure 10 is performed, relating each element to the measurements derived from the Sy sensor, which are [0,$dy_{2}$,−$dz_{2}$], giving rise to a second system of three equations with the same six unknowns as in the previous case.

In such virtue and knowing the measurements of the sensors, the mathematics of each sensor (Equations (Equation 22) and (Equation 23)) can be associated with the measurement of the two sensors forming the following vector [$dx_{1}$,0,−$dz_{1}$,0,$dy_{2}$,−$dz_{2}$]. Subsequently, a total set of six equations with six unknowns [$\theta_{1}$,$\theta_{2}$,$l_{1}$,$l_{2}$,$l_{3}$,$l_{4}$] can be structured in Equation (Equation 24), which represents the inverse kinematics approach as a method to determine the orientation angles $\theta_{1}$ and $\theta_{2}$ of the pendulum.(24)$$\left[\begin{array}{c} l_{p}\;cos\left(\theta_{2}+\frac{pi}{2}\right)+l_{2}\;sin\left(\theta_{2}+\frac{pi}{2}\right)\\ l_{2}\;cos\left(\theta_{1}+\frac{pi}{2}\right)cos\left(\theta_{2}+\frac{pi}{2}\right)-l_{1}\;sin\left(\theta_{1}+\frac{pi}{2}\right)-l_{p}\;cos\left(\theta_{1}+\frac{pi}{2}\right)sin\left(\theta_{2}+\frac{pi}{2}\right)\\ l_{1}\;cos\left(\theta_{1}+\frac{pi}{2}\right)-l_{p}\;sin\left(\theta_{1}+\frac{pi}{2}\right)sin\left(\theta_{2}+\frac{pi}{2}\right)+l_{2}\;cos\left(\theta_{2}+\frac{pi}{2}\right)sin\left(\theta_{1}+\frac{pi}{2}\right)\\ l_{p}\;cos\left(\theta_{2}+\frac{pi}{2}\right)+l_{4}\;sin\left(\theta_{2}+\frac{pi}{2}\right)\\ l_{3}\;sin\left(\theta_{1}+\frac{pi}{2}\right)-l_{p}\;cos\left(\theta_{1}+\frac{pi}{2}\right)sin\left(\theta_{2}+\frac{pi}{2}\right)+l_{4}\;cos\left(\theta_{1}+\frac{pi}{2}\right)cos\left(\theta_{2}+\frac{pi}{2}\right)\\ l_{4}\;cos\left(\theta_{2}+\frac{pi}{2}\right)sin\left(\theta_{1}+\frac{pi}{2}\right)-l_{3}\;cos\left(\theta_{1}+\frac{pi}{2}\right)-l_{p}\;sin\left(\theta_{1}+\frac{pi}{2}\right)sin\left(\theta_{2}+\frac{pi}{2}\right) \end{array}\right]=\left[\begin{array}{c} dx_{1}\\ 0\\ -dz_{1}\\ 0\\ dy_{2}\\ -dz_{2} \end{array}\right],$$

Among the six variables in Equation (Equation 24), focus will be placed on the first two, which correspond to the absolute angles $\theta_{1}$ and $\theta_{2}$ relative to the fixed axes $X_{0}$ and $Y_{0}$. Referring to Figure 7, these angles enable the determination of the pendulum’s position components through Equations (Equation 25) to (Equation 27). These coordinates are equivalent to those described in Equations (Equation 1) to (Equation 3).(25)$$x=l\;sin\theta_{1},$$(26)$$y=l\;sin\theta_{2},$$(27)$$z=-l\;cos\theta_{1}cos\theta_{2},$$

## 5. Experimental Results

The constructed prototype is shown in Figure 11, which consists of a pendulum rod with a mass at its lower end. Additionally, the sensors Sx and Sy, corresponding to those in Figure 6, were installed at a distance of 0.02 m from the pendulum’s rotation point. These sensors were positioned at a 90-degree angle relative to each other, aligning with the *X* and *Y* axes, respectively. The sensors used are the infrared position sensors (model TCRT5000 by Arduino). For data acquisition, the NI USB-6212 data acquisition device from National Instruments was employed and interfaced with MATLAB for data processing.

The collected data were processed using Equations (Equation 25) to (Equation 27), allowing for the calculation of each coordinate of the pendulum mass position.

The prototype measures 0.3 m in length, weighs 0.1 kg, and its circular plane of light reflection is 12 cm in diameter; this was located at a distance of −2 cm in the direction of the pendulum bar; the material used was of solid and light characteristic (for the case was cardboard paper), so as not to significantly affect the dynamics of the pendulum. Since the pendulum will have an equivalent oscillation period of $T=2\pi \sqrt{\left(l/g\right)}$ equivalent to 1.09 s, a sampling rate of 20 samples per second was adopted. Additionally, it was considered essential to include a filter in the signal acquired from the sensor to minimize noise. The filter used corresponds to the FIR filter coefficient of order 5, which yielded a very acceptable result in terms of noise cancellation for the signal from the TCRT5000 sensor. The code of the developed program is shown in Appendix A.

Two experimental cases are considered, corresponding to the cases developed in the simulations. Case 1 contemplates an initial displacement in the angle θ (Figure 1), which generates an oscillatory motion of the pendulum. Case 2 includes, in addition to an initial angle θ, an angular velocity in the other rotational axis corresponding to ϕ, aiming to produce a concentric helical response.

### 5.1. Experimental Results Case 1

For an initial condition [θ, ϕ, ω, ψ] = [10°,45°,0 rad/s,0 rad/s], the components generating the oscillatory motion along a diagonal trajectory between the *X* and *Y* axes have been calculated, where the amplitudes in the *X* and *Y* axes vary according to a sinusoidal function. Correspondingly, the amplitude along the *Z* axis shows an oscillatory negative displacement. These effects are observable in Figure 12a, Figure 12b and Figure 12c, respectively. For a better spatial appreciation of the pendulum trajectory, Figure 13 is presented.

Observing Figure 13a,b, it can be noted that the pendulum produces displacement along both the *X* and *Y* axes in a sinusoidal pattern. Given the initial condition of 45 degrees between the XY plane, these oscillations maintain a phase relationship. Finally, Figure 13c shows the upward and downward oscillation occurring at the pendulum’s tip, with amplitude directly related to the pendulum rod length. It is worth highlighting that the minimum height point corresponds to the pendulum’s length along the negative *Z* direction.

It is worth noting that sensor samples were collected over 10 s of experimentation. Longer simulation times occasionally caused the pendulum to exhibit an elliptical motion tendency. This effect is likely influenced by factors such as medium damping and possible friction at the oscillation pivot point, among others.

Since this prototype aims to verify the dynamic behavior and compare it with the simulated mathematical model, as well as validate the measurement method, the implemented system adequately achieves the study’s objectives. However, for practical applications, efforts should be increased in design aspects, including precise sensor placement Sx and Sy and proper data processing, especially concerning acquisition systems, signal noise, and offset filtering techniques.

Another visible aspect during experimentation is the minor influence of the damping factor; the signal appears to be preserved over time. To practically quantify the damping factor β, an extended experiment was conducted to estimate the amplitude decay in real oscillation measurements. Based on this behavior, a β factor that, when applied to Equations (Equation 14) and (Equation 15), produces a similar result was sought. The determined value of β was 0.01 based on experimental calibration. Since determining this factor is not the main objective of this research, this topic is not further emphasized.

### 5.2. Experimental Results Case 2

An initial condition is adopted to produce a conical pendulum motion. Thus, the initial condition [θ, ϕ, ω, ψ] = [10°, 45°, 0 rad/s, 2 rad/s], implies that with an initial angular velocity of $\dot{\phi }$ = 2 rad/s, a conical motion is generated (Figure 14 illustrates the initial instant of the movement). To fulfill this requirement, a mechanism based on a spring that has been adjusted to print a tangential linear velocity at the instant that includes a theta angle of 45 degrees was used (Figure 14). The required linear velocity corresponds to the value resulting from the product of ω and its perpendicular distance to the vertical axis of rotation, which is equivalent to a radius of r=l∗sin(45). This implies a velocity of 2(0.3sin(45)), corresponding to approximately 0.42 m/s. It should be emphasized that this experiment aims to capture, using the sensors, an approximate movement similar to the simulations in case two and, in this way, to record the sensor readings for later analysis.

Consequently, the mass position behavior in the *X*, *Y*, and *Z* axes changes following a sinusoidal pattern. These effects can be observed in Figure 15a, Figure 15b and Figure 15c respectively. Similarly to the previous case, Figure 16 is shown to provide a better spatial appreciation of the pendulum’s trajectory.

The results shown in Figure 15a,b reveal that the amplitudes differ slightly, indicating that the motion at the pendulum’s tip is not perfectly circular but rather exhibits a slight elliptical tendency. This effect can be attributed to the manual application of velocity used to generate the conical motion required for data acquisition. Considering that the pendulum’s tip motion is elliptical rather than circular, this implies an oscillatory behavior in the vertical component of the pendulum’s tip position, which can be observed in Figure 15c.

Observing Figure 16a, which shows a lateral view, the pendulum’s behavior in the YZ plane reveals a slight decrease in oscillation amplitude due to the natural damping of the surrounding medium. Regarding Figure 16b, representing a top view of the XY plane, it can be highlighted that the oscillation develops as circular with an elliptical tendency; this behavior results from the inclusion of the initial transverse angular velocity $\dot{\phi }$. Finally, Figure 16c presents a spatial perspective of the pendulum’s motion, where small undulations in some sections of the trajectory can be observed, caused by the motion not being purely circular and by the gradual loss of height. Based on Equation (Equation 3), the dependence of the position on the *Z*-axis of the pendulum mass is a direct function of $\theta_{1}$ and $\theta_{2}$; therefore, the elliptical behavior is seen from a higher perspective (Figure 16b); this implies generating a result with a wavelike behavior in the *Z*-coordinate of the system position, which causes the pendulum to tend towards the stability point in the lower part in the direction of negative *Z*, corresponding to a vertical arrangement without motion.

## 6. Discussions

The present investigation has demonstrated that it is possible to estimate the angular orientation of a spatial pendulum by a non-invasive optical sensing method, using orthogonally arranged infrared distance sensors bouncing off an auxiliary plane fixed orthogonally to the pendulum bar. This approach, which avoids physical contact with the oscillating system, is consistent with modern approaches described in the literature [11,12,13], where the need to minimize disturbances induced by measurement mechanisms in nonlinear and sensitive dynamic systems, such as the spherical pendulum, is emphasized.

This research builds upon previous works that utilize mechanical sensors or sensors coupled to the rotation axis [5,6], offering an alternative that minimizes friction and wiring, aspects that have historically impacted the fidelity of the measured dynamic behavior. As indicated by [14,15], optical sensors are suitable for high-sensitivity systems, and the results presented here reinforce this statement by showing that the signals obtained reflect the three-dimensional oscillatory pattern of the pendulum with reasonable accuracy.

The orientation reconstruction strategy, using the Denavit–Hartenberg model, which is widely employed in robotic kinematics [19,20,21,22], proved effective in this case, as it enables the transformation of linear measurements into rotation angles. This aligns with research that applies matrix methods for modeling and control of multibody mechanisms, as shown in [25,26].

On the other hand, the implemented dynamic model incorporated a generalized damping term that lumps real physical effects, such as friction and dissipation in the environment. The inclusion of this factor, although empirically estimated (β≈0.01), is in agreement with studies that highlight the importance of considering such losses to achieve more realistic simulations [18,23,24]. The ability to adjust this parameter after prolonged observations of amplitude decay was key to achieving a proper match between experimental data and simulation. However, some limitations were identified as consistent with previous observations in the literature [27,28,29]. The accuracy of the TCRT5000 sensors, although sufficient to validate the approach, can be improved by using industrial-grade optical sensors. Additionally, as noted by [17,18], digital signal filtering and proper data processing are crucial for obtaining robust measurements. Therefore, the use of more sophisticated filtering techniques (e.g., Kalman or weighted average) is a logical next step for future work.

Finally, although validation of the measurement model was achieved, opportunities for improvement in the design of the acquisition system are recognized, particularly in the exact location of the Sx and Sy sensors. This aspect, already discussed in [3,4], also directly impacts the quality of the result, especially when observing patterns such as elliptical trajectories induced by uncontrolled initial conditions.

## 7. Conclusions

The models describing the dynamic behavior of the spatial pendulum have been developed in the literature. However, when its motion is experimentally observed, several factors arise that cause the oscillation amplitude to decrease over time. Therefore, it is essential to incorporate into the model a damping factor that accounts for all relevant effects, such as medium density, pendulum geometry, and friction at the pivot point, among others. This damping factor can be approximately estimated through prolonged experimentation, where the decay of oscillation amplitude over time is recorded. This value can then be incorporated into the equations of motion, validated, and adjusted in simulations to ensure that the oscillation response under a given scenario closely matches the actual behavior.

The sensors are used to measure distance, providing an analog signal proportional to the distance. This signal must be digitally processed using noise filters to achieve a more reliable measurement. Despite this, the sensors proved helpful in this investigation. Being a main component, these could be improved with better performance sensors for industrial applications, considering that the TCRT5000 sensor has a limitation in terms of the detection reflection distance that goes from 1 mm to 25 mm, this implies that the reflection plane must be positioned at a distance less than 25 mm; therefore, those oscillations that exceed this distance will present inconsistencies.

After obtaining the light reflection measurement on the disk, it is necessary to infer the corresponding pendulum rotation angles. A matrix-based procedure was employed for this calculation, using the Denavit–Hartenberg method, which avoids making geometric deductions and instead adopts a matrix and vector technique based on the concept of homogeneous matrices. This involved defining segments generated on the plane and the sensor measurement through direct kinematics, resulting in a system of equations whose unknowns are the rotation angles on each axis. Subsequently, by solving the inverse kinematics problem, the rotation angles of the pendulum can be obtained. Experimental results were corroborated with simulations. Regarding initial conditions for the pendulum’s motion, accuracy can be improved for future purposes. In this study, the goal was to demonstrate that the sensing technique is functional, so exact initial conditions were not emphasized. Observing similarity and motion trends sufficed to conclude that the method is valid and can be refined in various aspects, including the generation of initial conditions for experiments and simulations.

Additionally, it is essential to highlight that, as a prototype intended to verify dynamic behavior and compare it with the simulated mathematical model, the implemented system successfully meets the study’s objectives by validating the measurement method. Nevertheless, for practical application, it is clear that further efforts must be directed toward design improvements, such as the precise positioning of sensors Sx and Sy and the proper processing of their data, including acquisition systems and noise filtering, and signal offset correction techniques.

Another noticeable aspect during experimentation is the minimal influence of the damping factor; the signal persists over time. To quantify the damping factor β practically, prolonged experimentation was performed. This allowed estimation of the amplitude decay in the real oscillation measurements. Considering this behavior, a value of β=0.01 was adopted, which in simulations produces results closely matching the experimental responses. It is essential to note that the adoption of factor B will enable the observation of the damping tendency of the medium in which the movement develops. It is essential to note that the adoption of factor β will enable the observation of the damping tendency of the medium in which the movement develops. This aspect is not of primary interest. However, it is necessary to include it in the mathematical model for simulation purposes.

Finally, it is worth emphasizing that the exact difference between the experiments and the simulations that incorporate the theoretical calculation has not been quantified, as the generation of the initial conditions was not the primary purpose of this research. However, it is essential that the tendency of movement for each case remains dependent and allows for validation of the proposed sensing method. Although the effectiveness of the method can be verified by statically positioning the pendulum at different points, it is a very elementary approach; therefore, its validation in a moving environment and with the dynamics of the mechanism itself is considered enriching. In this sense, a new window of research is opened involving the performance of the pendulum system and its motion control.

## Figures and Tables

**Figure 1 sensors-25-04624-f001:**
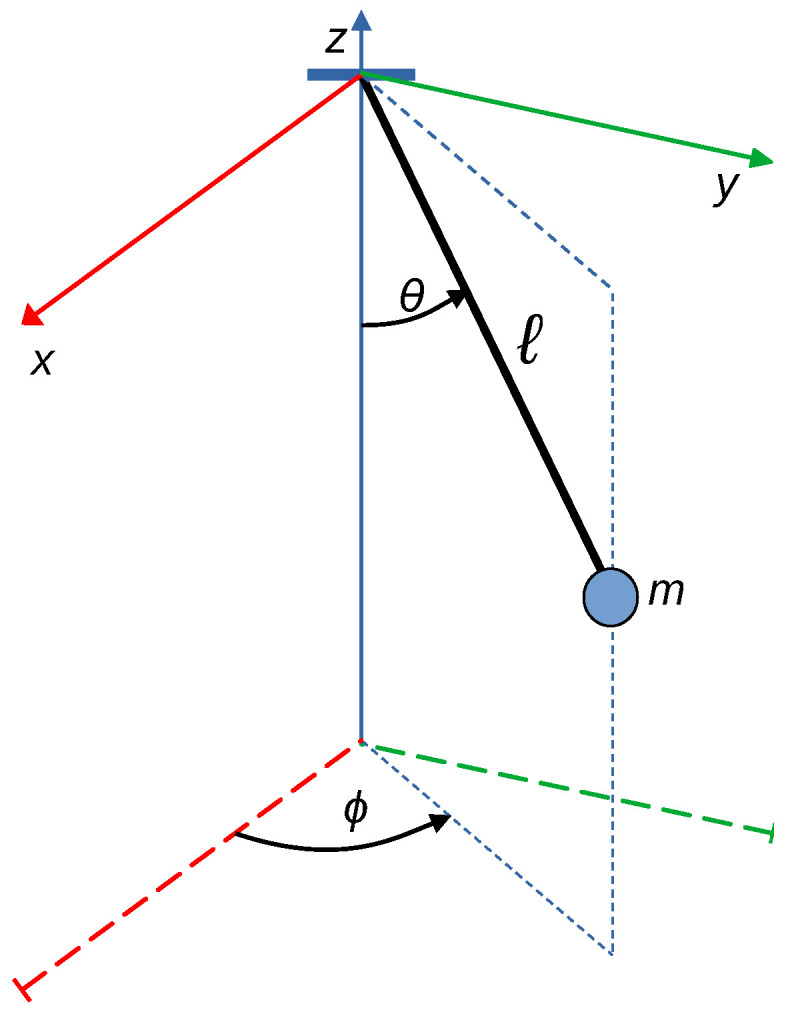
Spatial pendulum, representation of variables identifying the motion.

**Figure 2 sensors-25-04624-f002:**
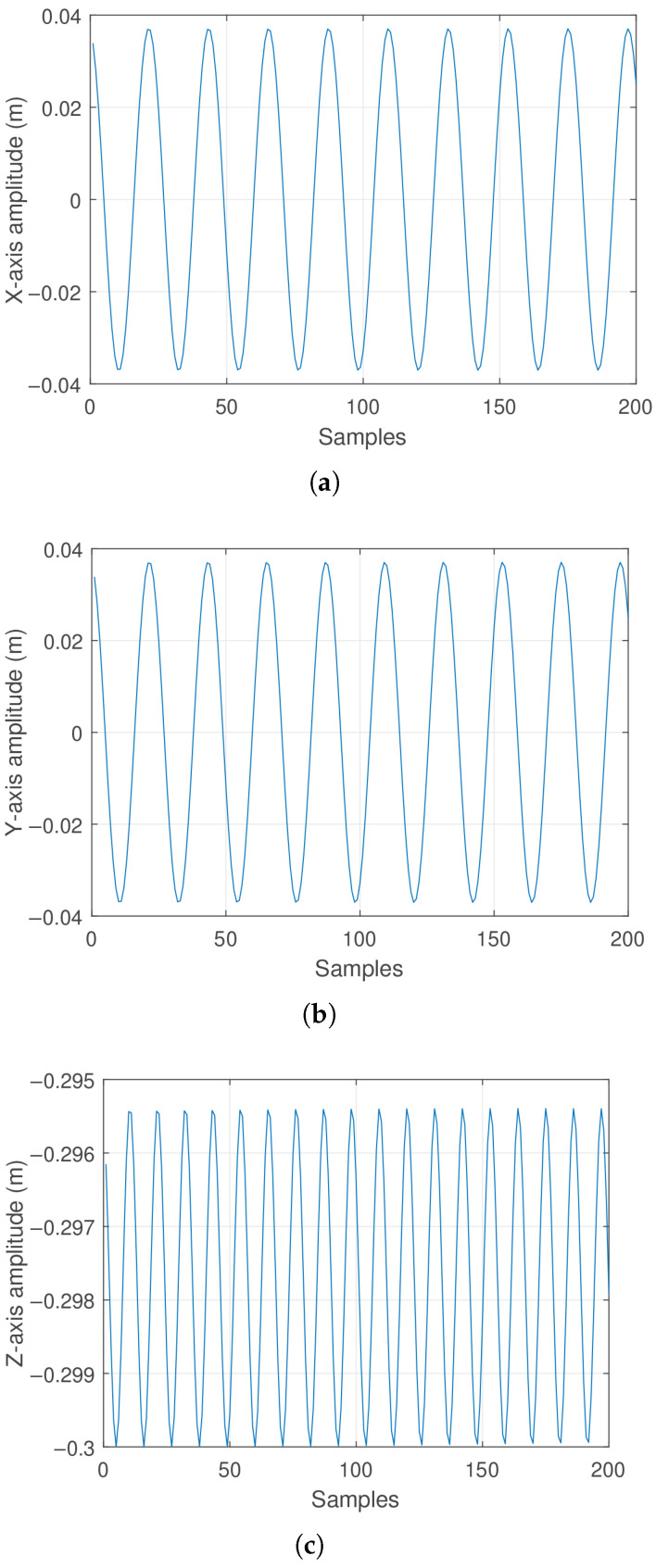
Pendulum oscillation response: (**a**) Displacement along the X-axis. (**b**) Displacement along the Y-axis. (**c**) Displacement along the Z-axis.

**Figure 3 sensors-25-04624-f003:**
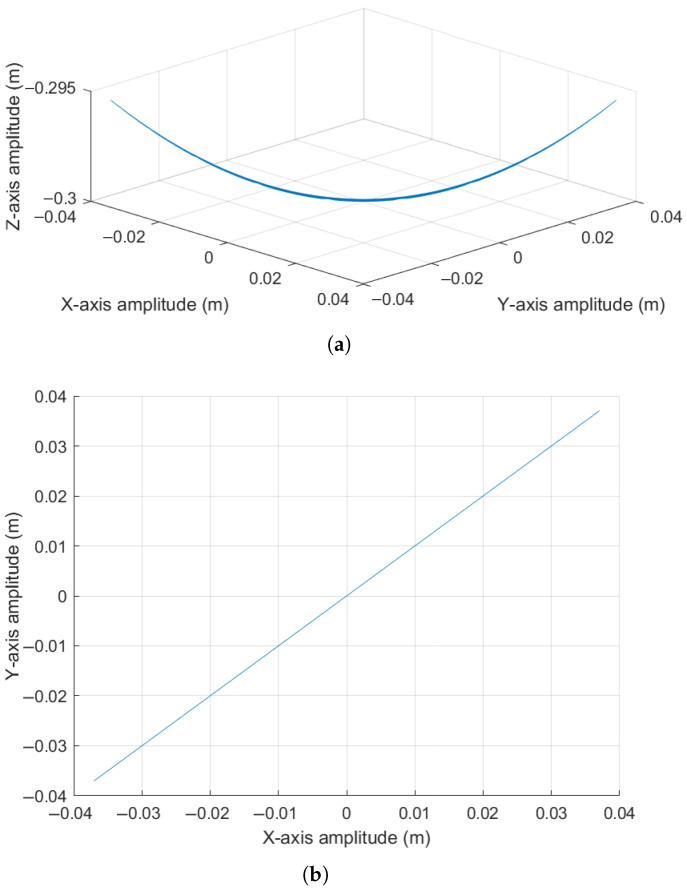
Pendulum oscillation response: (**a**) Spatial oscillation behavior trajectory. (**b**) Oscillation trajectory viewed from above in the XY plane.

**Figure 4 sensors-25-04624-f004:**
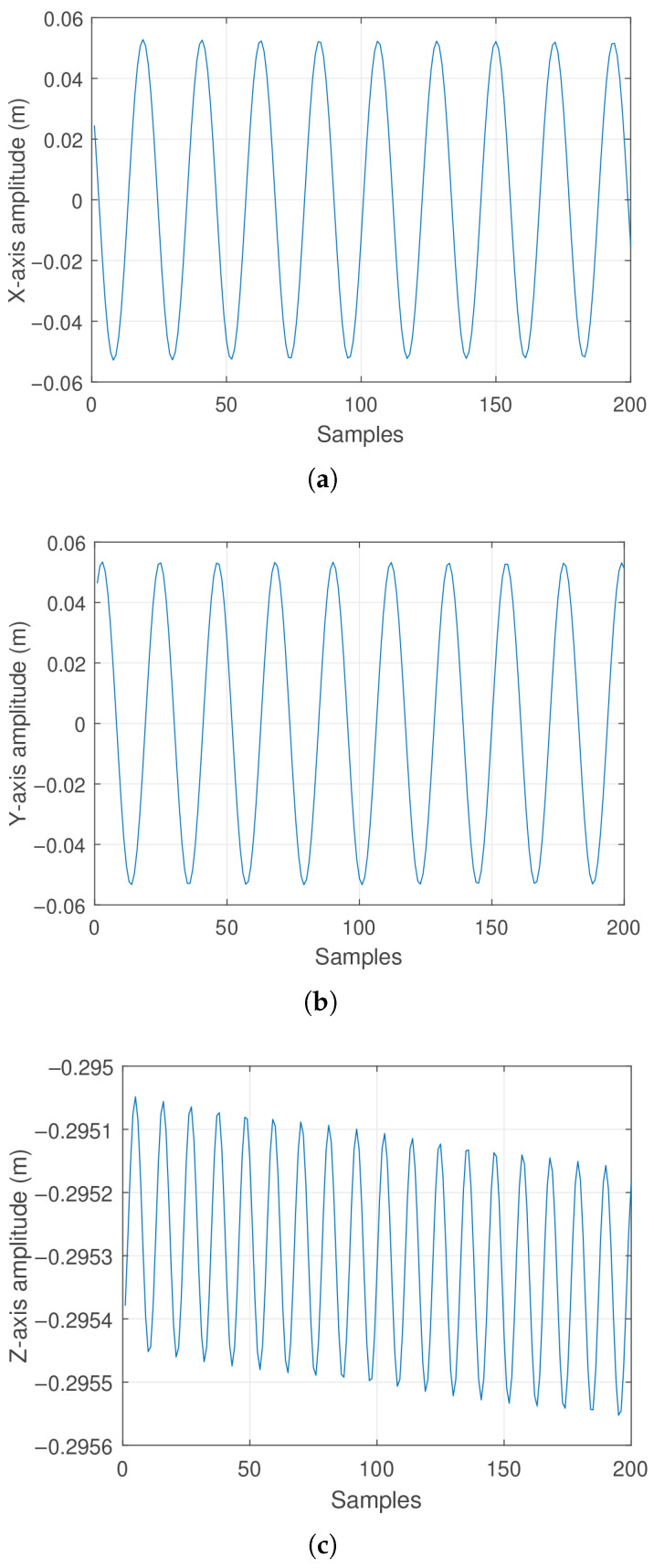
Pendulum oscillation response considering a non-zero initial angular velocity $\dot{\phi }$: (**a**) Displacement along the *X*-axis. (**b**) Displacement along the *Y*-axis. (**c**) Displacement along the *Z*-axis.

**Figure 5 sensors-25-04624-f005:**
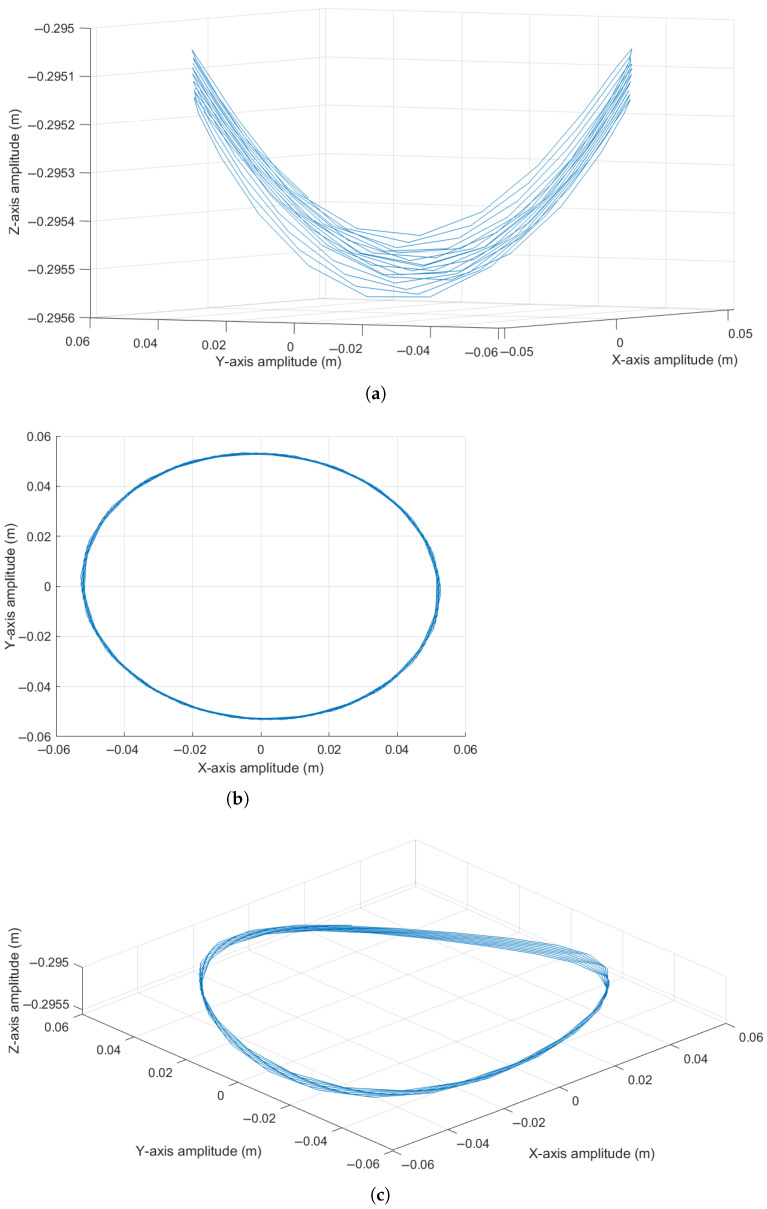
Pendulum oscillation response considering a non-zero initial angular velocity $\dot{\phi }$: (**a**) Side view in the XZ plane. (**b**) Top view in the XY plane. (**c**) Spatial view.

**Figure 6 sensors-25-04624-f006:**
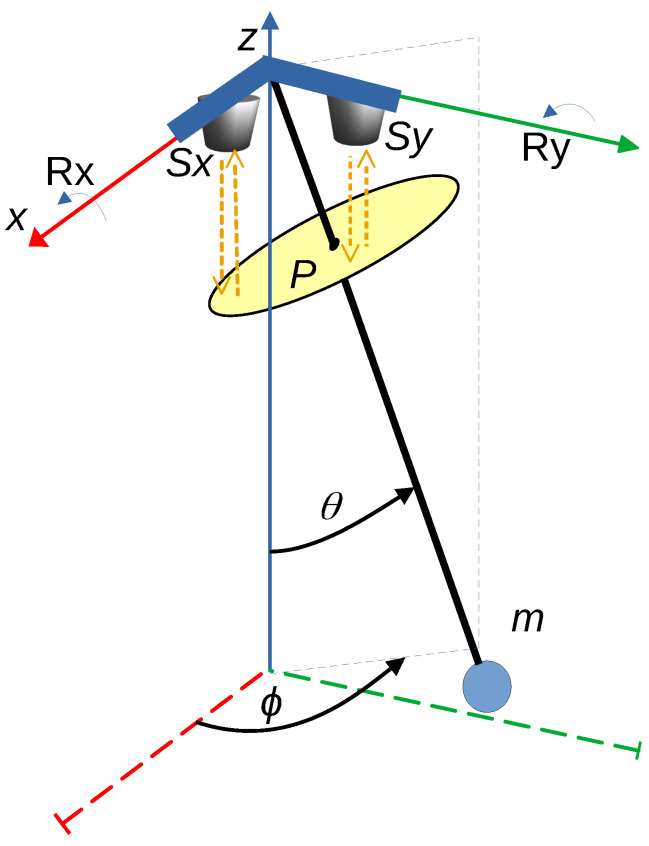
Pendulum system and integration of the sensing mechanism.

**Figure 7 sensors-25-04624-f007:**
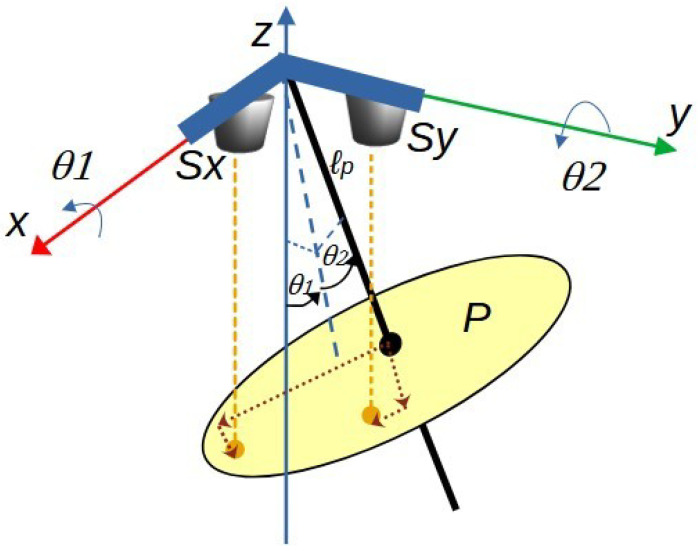
Distance sensing system based on light reflection on plane P, relative to absolute rotation about the X and Y axes.

**Figure 8 sensors-25-04624-f008:**
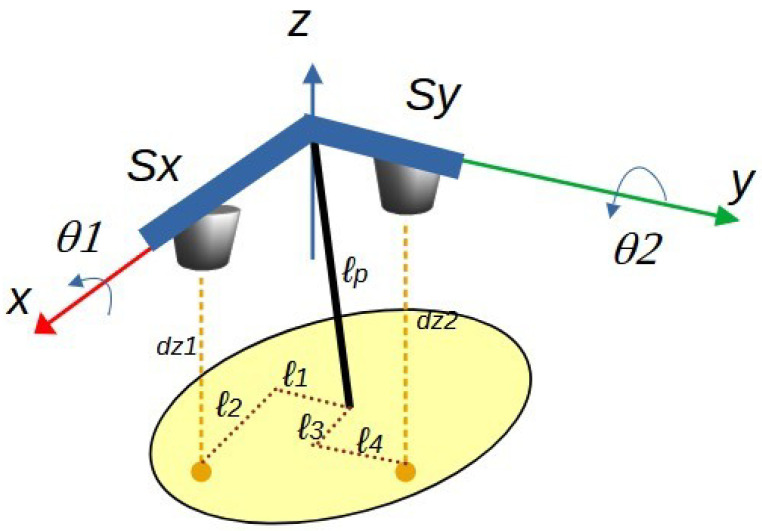
Reflection distance sensing system: segments defining the distance measurements of each sensor.

**Figure 9 sensors-25-04624-f009:**
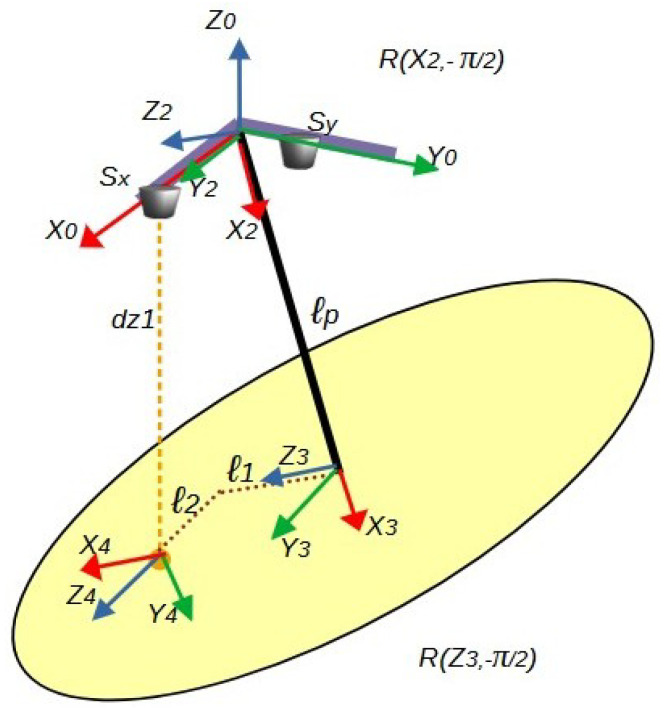
Segments that define the distance measured by sensor Sx, along with the coordinate frame assignments required for the computation of forward and inverse kinematics.

**Figure 10 sensors-25-04624-f010:**
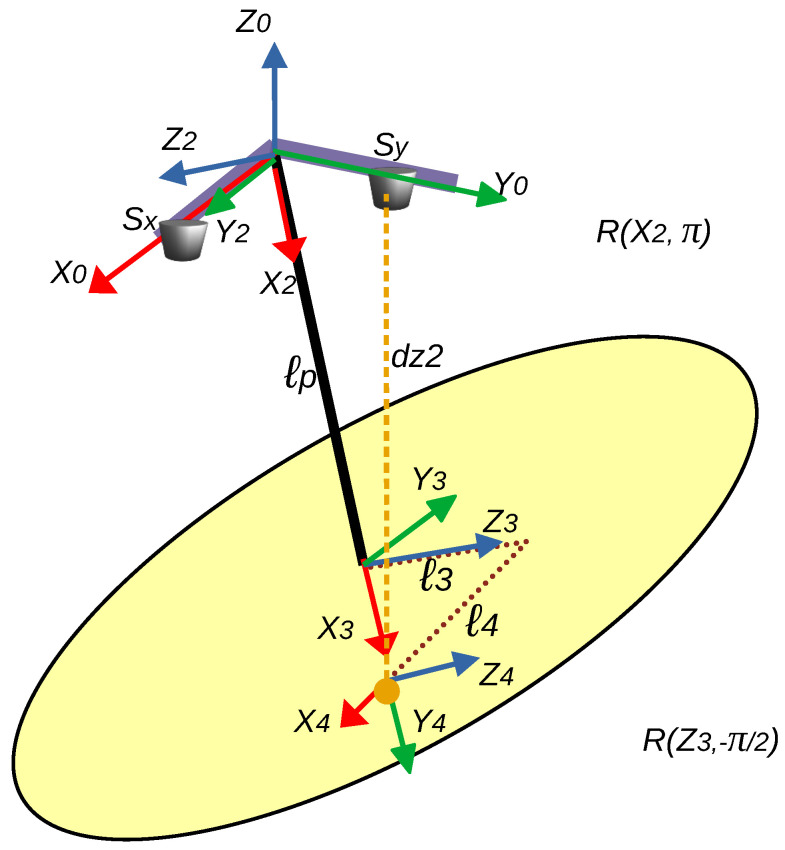
Segments generating the distance measured by sensor Sy, coordinate system for the calculation of direct and inverse kinematics.

**Figure 11 sensors-25-04624-f011:**
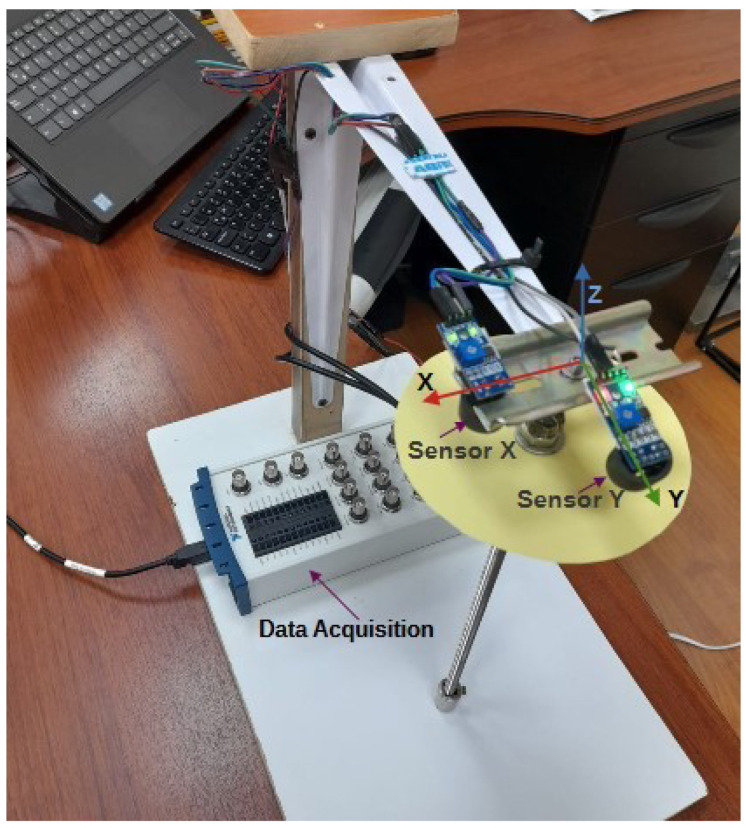
Spatial pendulum prototype.

**Figure 12 sensors-25-04624-f012:**
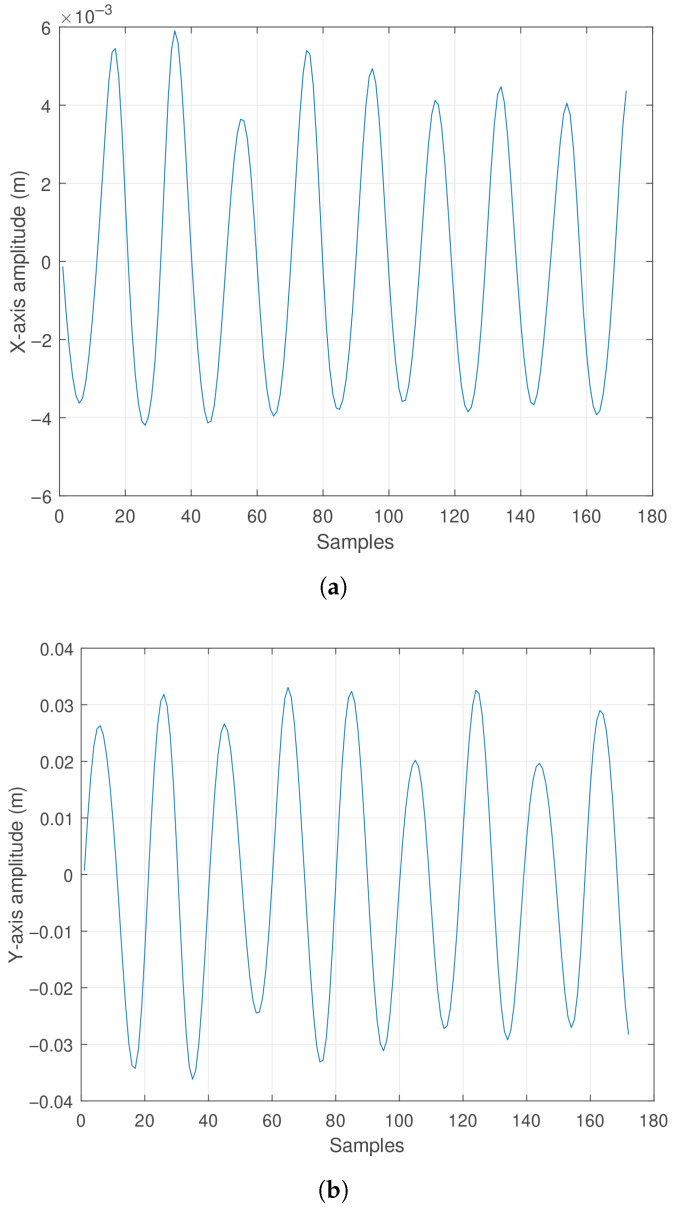
Actual oscillation response of the pendulum: (**a**) Displacement along the *X* axis. (**b**) Displacement along the *Y* axis. (**c**) Displacement along the *Z* axis.

**Figure 13 sensors-25-04624-f013:**
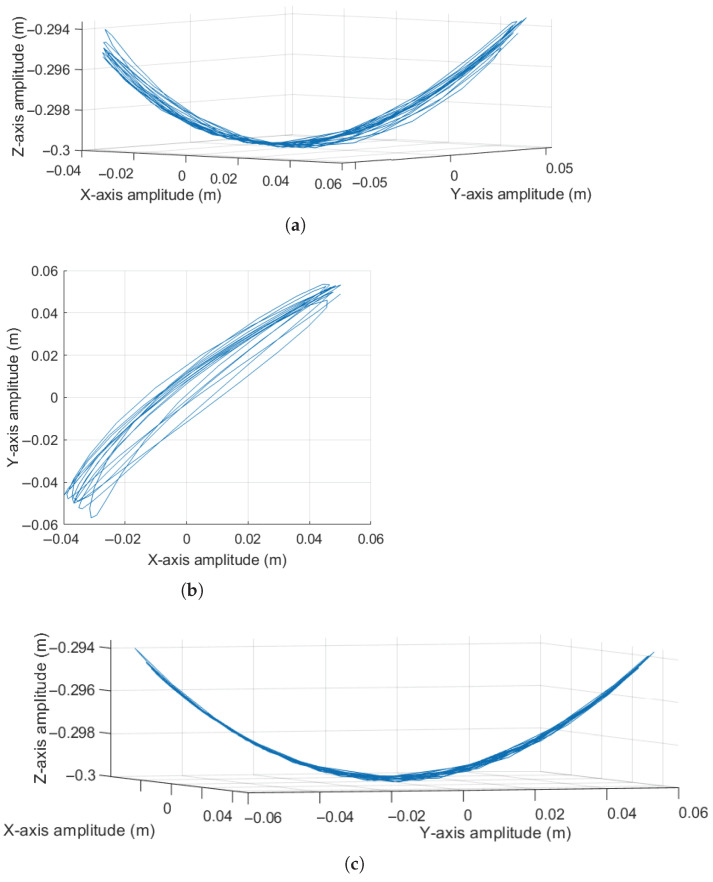
Real oscillation response of the pendulum: (**a**) Lateral view in the XZ plane. (**b**) Top view in the XY plane. (**c**) Three-dimensional view.

**Figure 14 sensors-25-04624-f014:**
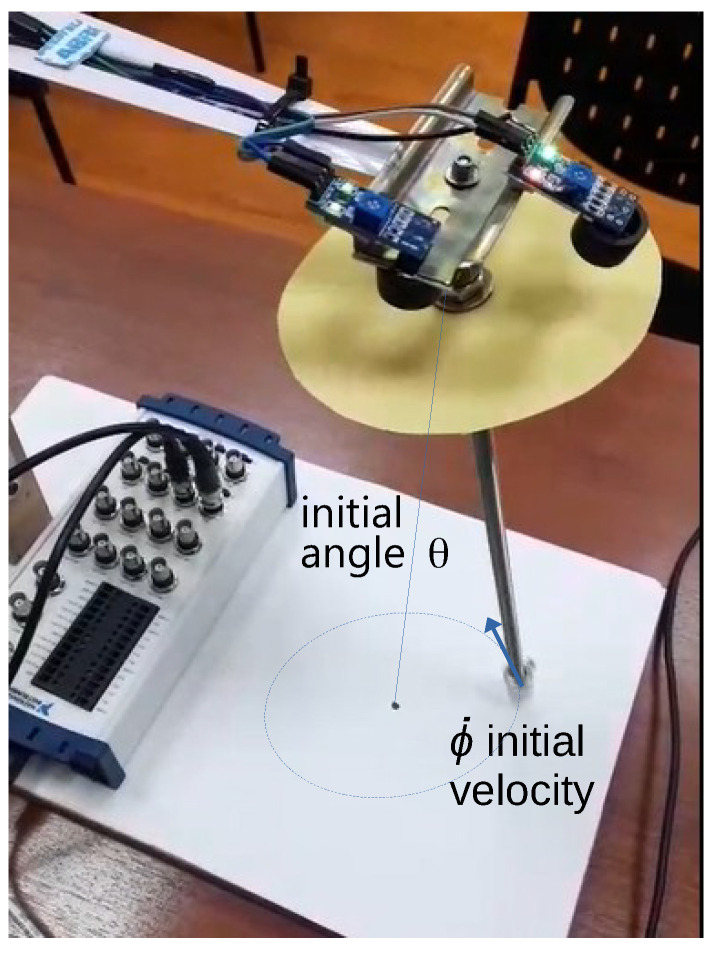
Initial motion condition of the pendulum.

**Figure 15 sensors-25-04624-f015:**
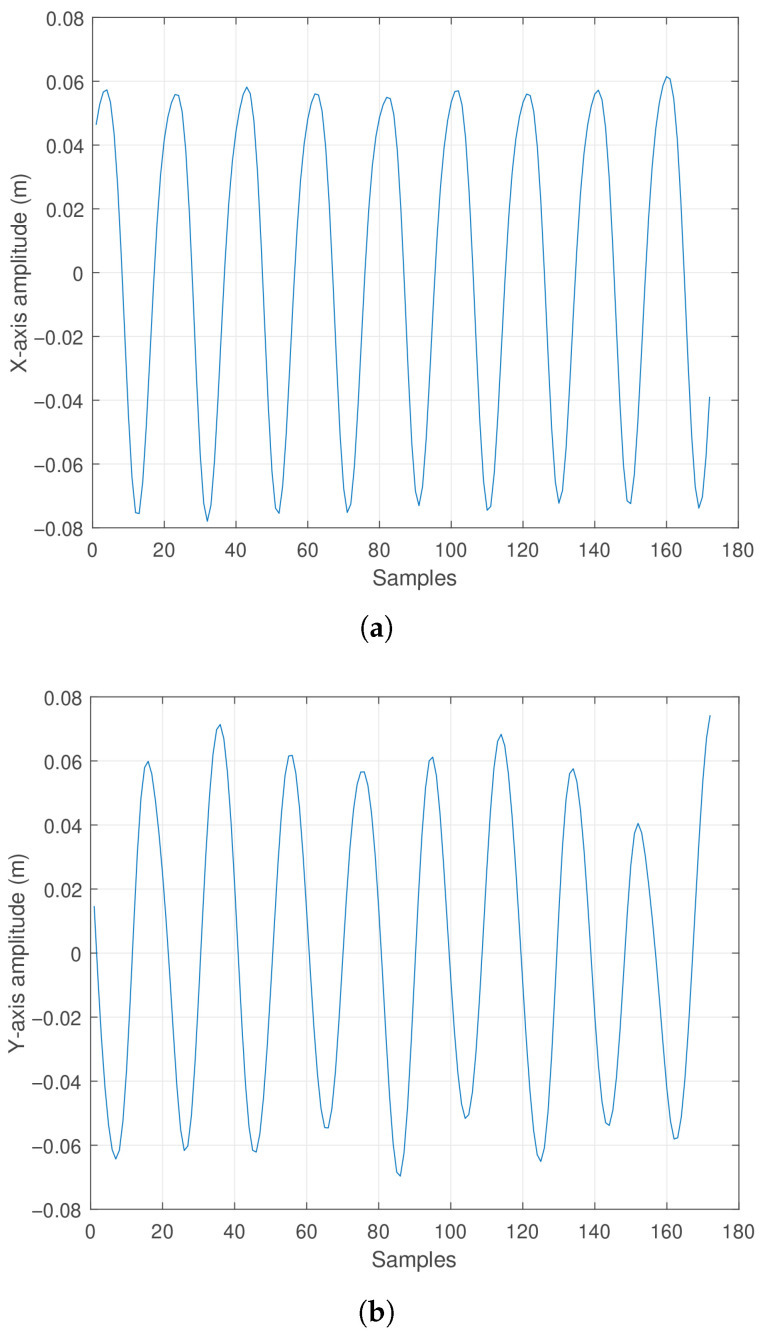
Real oscillation response of the pendulum considering a non-zero initial angular velocity $\dot{\phi }$: (**a**) Displacement along the *X*-axis. (**b**) displacement along the *Y*-axis. (**c**) Displacement along the *Z*-axis.

**Figure 16 sensors-25-04624-f016:**
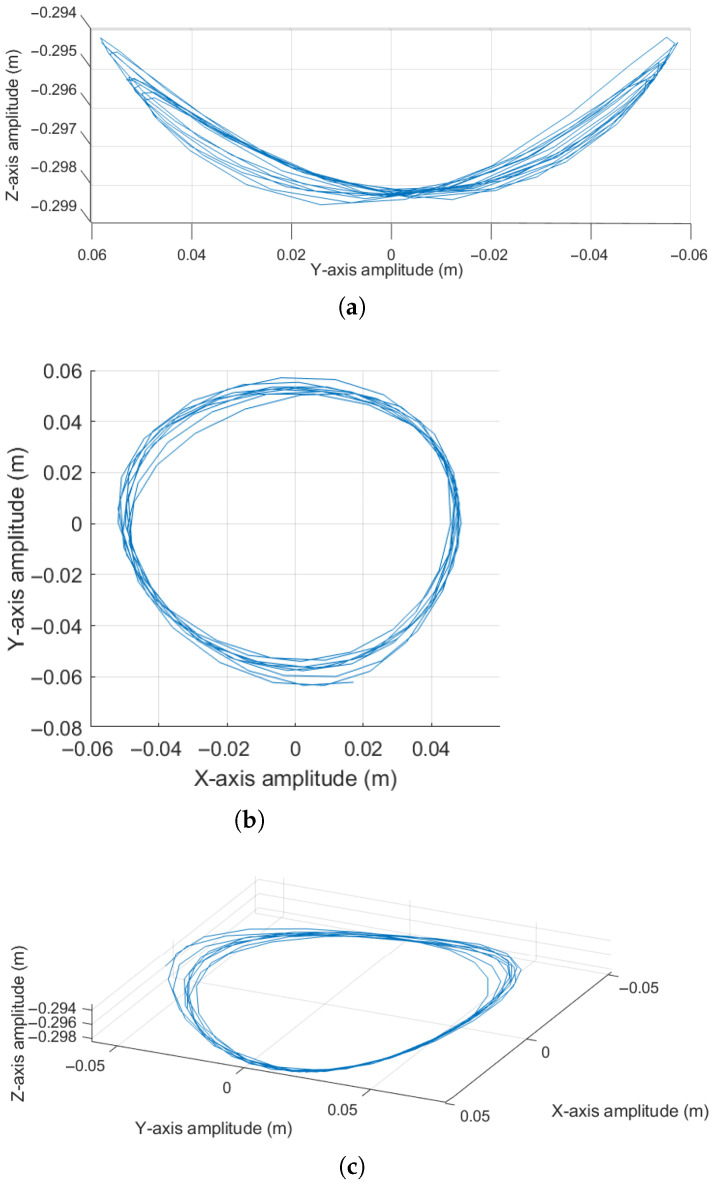
Real oscillation response of the pendulum considering a non-zero initial angular velocity $\dot{\phi }=\psi$: (**a**) Lateral view in the XZ plane. (**b**) Top view in the XY plane. (**c**) Three-dimensional view.

**Table 1 sensors-25-04624-t001:** Denavit–Hartenberg parameters relating to the distance measured by sensor Sx.

Matrix	θ	*d*	*a*	α
A10	0	0	0	π/2 + $\theta_{1}$
A21	−π/2 + $\theta_{2}$	0	0	0
A32	0	0	$l_{p}$	0
A43	π/2	$l_{1}$	$l_{2}$	0

**Table 2 sensors-25-04624-t002:** Denavit–Hartenberg parameters relating to the distance measured by sensor Sy.

Matrix	θ	*d*	*a*	α
A10	0	0	0	π/2 + $\theta_{1}$
A21	−π/2 + $\theta_{2}$	0	0	0
A32	0	0	$l_{p}$	π
A43	−π/2	$l_{3}$	$l_{4}$	0

## Data Availability

Data are contained within the article.

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
