# Peer review of "Non-Invasive Position Measurement of a Spatial Pendulum Using Infrared Distance Sensors"

_sensors, 2025, doi:10.3390/s25154624_

Round 1
Reviewer 1 Report
Comments and Suggestions for Authors
In this research paper(sensors-3727424-peer-review-v1), this work presents the modeling and sensing of the spherical pendulum, integrating a novel non-invasive measurement scheme based on infrared sensors arranged in a quadrature configuration. I would like to recommend this manuscript to be published in sensors after minor revisions.
Some detail information needs to be clarified:
- The introduction somewhat vaguely addresses the limitations of existing technologies and the advantages of the new method. It is recommended to provide a more detailed comparison of the differences between traditional measurement methods (such as potentiometers and encoders) and the infrared sensor solution in terms of accuracy, invasiveness, and long-term stability to better highlight the innovation and necessity of the proposed method.
- The connection between the presentation and discussion of experimental results is not tight enough. For example, when comparing simulation and experimental results, it is not clearly pointed out which differences are caused by sensor errors and which are due to inaccurate initial conditions of the experiment.
- On Page 1, Line6, "leveraging light reflection in a perpendicular plane aligned with the pendulum bar." If the intended meaning is to utilize the reflection of light on a vertical plane aligned with the pendulum bar, then the more appropriate preposition is "on," as this emphasizes that the reflection occurs on the plane itself.
- The experimental data sampling rate and data processing algorithms (such as the type and parameters of filtering algorithms) are not specified in detail, which may affect the credibility and reproducibility of the experimental results. These details need to be added in the experimental methods section.
- On Page18, Line 305, The suggestion is to change the phrase "the most acceptable value found was β = 0.01" to "the determined value of β was 0.01 based on experimental calibration."
Author Response
Dear Reviewer, we, the authors, commend your contribution through the recommendations issued by you, which undoubtedly enrich our research and motivate us to continue.
Each observation has been analyzed, and we have worked to give it the best attention. In addition, the quality of the images has been improved, and the translation has been revised. We report in detail the adjustments made to the document as follows:
- The introduction somewhat vaguely addresses the limitations of existing technologies and the advantages of the new method. It is recommended to provide a more detailed comparison of the differences between traditional measurement methods (such as potentiometers and encoders) and the infrared sensor solution in terms of accuracy, invasiveness, and long-term stability to better highlight the innovation and necessity of the proposed method.
The introduction has been strengthened from page 1, line 25 to page 2, line 34, emphasizing the differences between traditional measurement methods (such as potentiometers and encoders) and the infrared sensor solution in terms of accuracy, invasiveness, and long-term stability, the text included required the addition of five bibliographic sources, the text added is:
“Specifically, methods incorporating potentiometers and encoders have been widely used due to their simplicity and high accuracy in control and monitoring applications [7], [8]. These devices present significant limitations in terms of invasiveness and long-term stability. In particular, potentiometers require direct physical contact, which makes them invasive solutions, especially problematic in biomedical or high-sensitivity applications, where contact may alter the system or compromise its integrity [9], [10]. Additionally, both potentiometers and encoders are susceptible to mechanical wear, contaminant accumulation, and temperature fluctuations, which can impact accuracy and necessitate periodic maintenance or calibration to maintain optimal performance [8].”
In addition, another text justifying the method was included on page 2 between lines 41 to 46, as follows:
“These sensors, especially those based on infrared spectroscopy or proximity beams, have demonstrated high accuracy comparable to that of traditional methods, with the advantage of eliminating the physical wear and tear inherent to contact systems. However, their long-term stability can be affected by instrumental drift and environmental conditions, so the implementation of online calibration and dynamic compensation techniques, such as orthogonal projection, becomes essential to maintain their reliability in special applications [10], [16].”
The titles of the references included are:
- A conceptual development of novel ultra-precision dimensional measurement technology
- In Situ Drift Monitoring and Calibration of Field-Deployed Potentiometric Sensors Using Temperature Supervision
- On-line recalibration of spectral measurements using metabolite injections and dynamic orthogonal projection
- Invasive and non-invasive measurement in medicine and biology: Calibration issues
- The Good pH probe: non-invasive pH in-line monitoring using Good buffers and Raman spectroscopy
- The connection between the presentation and discussion of experimental results is not tight enough. For example, when comparing simulation and experimental results, it is not clearly pointed out which differences are caused by sensor errors and which are due to inaccurate initial conditions of the experiment.
To clarify the connection between the simulations and the experimental results, it was emphasized that the method could be validated only by statically positioning the pendulum system at different angles. This aspect was pre-executed, but it was thought to be scarce, which is why the idea of generating the dynamics itself by evaluating the trend was adopted, since the initial conditions, especially the printing velocities, need greater treatment in what is being worked on in a new work involving the control system. In this sense, a paragraph was included that summarizes what is described on page 22, lines 463 to 471, being:
“Finally, it is worth emphasizing that the exact difference between the experiments and the simulations that incorporate the theoretical calculation has not been quantified, as the generation of the initial conditions was not the primary purpose of this research. However, it is essential that the tendency of movement for each case remains dependent and allows for validation of the proposed sensing method. Although the effectiveness of the method can be verified by statically positioning the pendulum at different points, it is a very elementary approach; therefore, its validation in a moving environment and with the dynamics of the mechanism itself is considered enriching. In this sense, a new window of research is opened involving the performance of the pendulum system and its motion control. “
- On Page 1, Line6, "leveraging light reflection in a perpendicular plane aligned with the pendulum bar." If the intended meaning is to utilize the reflection of light on a vertical plane aligned with the pendulum bar, then the more appropriate preposition is "on," as this emphasizes that the reflection occurs on the plane itself.
The observation is enriching for the interpretation of the idea; therefore, the recommendation is accepted, and the preposition “in” is changed to “on” on page 1, line 6.
- The experimental data sampling rate and data processing algorithms (such as the type and parameters of filtering algorithms) are not specified in detail, which may affect the credibility and reproducibility of the experimental results. These details need to be added in the experimental methods section.
A text is inserted detailing the value of the sampling frequency chosen as a function of the natural period of the pendulum, in addition to detailing the characteristics of the filter used for the treatment of the signal acquired from the sensor, and Appendix A is added containing the MATLAB code developed and that could be useful when recreating the experimentation.
The text included in page 15, line 295 to 304 is:
“The prototype measures 0.3m in length, weighs 0.1kg, and its circular plane of light reflection is 12cm in diameter; this has been located at a distance of -2cm in the direction of the pendulum bar; the material used was of solid and light characteristic (for the case was cardboard paper), so as not to significantly affect the dynamics of the pendulum. Since the pendulum will have an equivalent oscillation period of T = 2π √ (l/g) equivalent to 1.09 seconds, a sampling rate of 20 samples per second is adopted. Additionally, it has been considered essential to include a filter in the signal acquired from the sensor to minimize noise. The filter used corresponds to the FIR filter coefficient of order 5, which yielded a very acceptable result in terms of noise cancellation for the signal from the TCRT5000 sensor. The code of the developed program is shown in Appendix A.”
- On Page18, Line 305, The suggestion is to change the phrase "the most acceptable value found was β = 0.01" to "the determined value of β was 0.01 based on experimental calibration."
The timely comment for interpretation is considered; therefore, the text on page 15, line 338 is substituted, remaining:
“The determined value of β was 0.01 based on experimental calibration.”
Kind regards
Marco Carpio

Reviewer 2 Report
Comments and Suggestions for Authors
In this article, the modelling and sensing of the spherical pendulum is performed. For the sensing stage, a non-invasive measurement scheme based on infrared sensors arranged in a quadrature configuration is presented. For the measurements, the pendulum prototype was implemented. Simulations were performed based on the developed model.
The intention accordingly was to test the functionality of the sensing technique compared to the developed model (line 365, in the conclusions section). Although similar results, are presented, there are several points for improvement, many of which have been identified by the authors as future work:
- The way in which the manual application of the radial velocity is carried out is not accurate. Therefore, the authors must devise and implement ways to make it so.
- The initial conditions must be controlled.
- The error between experimental and theoretical results must be quantified.
- All measurements must be reliable. The use of noise filters, as has also been observed, should be applied.
The dimensions of the complete system and the materials used are not given, only the height and weight are mentioned, then it could not be exactly replicated.
Authors mentioned that Figure 16c presents a spatial perspective of the pendulum's motion, where small undulations in some sections of the trajectory can be observed caused by the motion not being purely circular and by the gradual loss of height. But it is necessary a further analysis of these causes, as it even appears to be a system that is not mechanically stable.
In addition, the experimental set-up must be provided.
In Figure 9, it is necessary to locate the frames, as well as the locations of Sx and Sy, which appear different from the case shown in Figures 7 and 8.
It is mentioned in the conclusions that to quantify the damping factor β practically, prolonged experimentation was performed. It is perceived throughout the reading that no repeated experiments were carried out. It is necessary to clarify this point. In case experiments have not been repeated, they should be repeated, and the relevant analysis of results should be carried out. The same applies to all experiments.
From my point of view, observing similarity and movement trends is not enough to conclude that the method is valid. The measurements made must have a high degree of accuracy before this work could be published.
Author Response
Dear Reviewer, we, the authors, commend your contribution through the recommendations issued by you, which undoubtedly enrich our research and motivate us to continue.
Each observation has been analyzed, and we have worked to give it the best attention. In addition, the quality of the images has been improved, and the translation has been revised. We report in detail the adjustments made to the document as follows:
- The way in which the manual application of the radial velocity is carried out is not accurate. Therefore, the authors must devise and implement ways to make it so.
- The initial conditions must be controlled.
The procedure for the generation of the conical motion involving the insertion of an initial tangential velocity accompanied by an initial inclination angle is detailed. On page 17, lines 345 to 352, the following explanation was included:
“To fulfill this requirement, a mechanism based on a spring that has been adjusted to print a tangential linear velocity at the instant that includes a theta angle of 45 degrees was used (Figure 14). The required linear velocity corresponds to the value resulting from the product of ω and its perpendicular distance to the vertical axis of rotation, which is equivalent to a radius of r = l ∗ sin(45). This implies a velocity of 2(0.3sin(45)), corresponding to approximately 0.42m/s. It should be emphasized that this experiment aims to capture, using the sensors, an approximate movement similar to the simulations in case two and, in this way, to record the sensor readings for later analysis.”
- The error between experimental and theoretical results must be quantified.
Considering that this aspect is part of future research in development that contemplates the control of the pendulum system, this requirement is justified with the purposes of the current research that intends to validate the measurement method using infrared sensors, a light reflection plane, and Denavit Hartenberg matrix mathematics. Therefore, two paragraphs were included in the conclusions section, the first on page 21, lines 431 to 433, being the following:
“using the Denavit-Hartenberg method, who avoids making geometric deductions and instead adopts a matrix and vector technique based on the concept of homogeneous matrices.”
The second paragraph was inserted on page 22, lines 463 to 471, being:
“Finally, it is worth emphasizing that the exact difference between the experiments and the simulations that incorporate the theoretical calculation has not been quantified, as the generation of the initial conditions was not the primary purpose of this research. However, it is essential that the tendency of movement for each case remains dependent and allows for validation of the proposed sensing method. Although the effectiveness of the method can be verified by statically positioning the pendulum at different points, it is a very elementary approach; therefore, its validation in a moving environment and with the dynamics of the mechanism itself is considered enriching. In this sense, a new window of research is opened involving the performance of the pendulum system and its motion control. “
- All measurements must be reliable. The use of noise filters, as has also been observed, should be applied.
- The dimensions of the complete system and the materials used are not given, only the height and weight are mentioned, then it could not be exactly replicated.
The two observations are considered, and a text is inserted detailing the value of the sampling frequency chosen as a function of the natural period of the pendulum, in addition to detailing the characteristics of the filter used for the treatment of the signal acquired from the sensor. Appendix A is added, containing the Matlab code developed, and that could be useful when recreating the experimentation.
The text included on page 15, lines 295 to 304, is:
“The prototype measures 0.3m in length, weighs 0.1kg, and its circular plane of light reflection is 12cm in diameter; this has been located at a distance of -2cm in the direction of the pendulum bar; the material used was of solid and light characteristic (for the case was cardboard paper), so as not to significantly affect the dynamics of the pendulum. Since the pendulum will have an equivalent oscillation period of T = 2π √ (l/g) equivalent to 1.09 seconds, a sampling rate of 20 samples per second is adopted. Additionally, it has been considered essential to include a filter in the signal acquired from the sensor to minimize noise. The filter used corresponds to the FIR filter coefficient of order 5, which yielded a very acceptable result in terms of noise cancellation for the signal from the TCRT5000 sensor. The code of the developed program is shown in Appendix A.”
- Authors mentioned that Figure 16c presents a spatial perspective of the pendulum's motion, where small undulations in some sections of the trajectory can be observed caused by the motion not being purely circular and by the gradual loss of height. But it is necessary a further analysis of these causes, as it even appears to be a system that is not mechanically stable.
Further explanatory analysis is made on the undulations present in the behavior of the Z coordinate of the pendulum position; the corresponding text is inserted on page 18, lines 371 to 376, being:
“Based on equation 3, the dependence of the position on the Z-axis of the pendulum mass is a direct function of θ1 and θ2; therefore, the elliptical behavior is seen from a higher perspective (Figure 16b); this implies generating a result with a wavelike behavior in the Z-coordinate of the system position, which causes the pendulum to tend towards the stability point in the lower part in the direction of negative Z, corresponding to a vertical arrangement without motion. “
- In addition, the experimental set-up must be provided.
In a way, this requirement is addressed with the paragraphs included regarding the initial conditions and the way in which the speed is manually applied; these paragraphs have been inserted on page 15, lines 295 to 304:
“The prototype measures 0.3m in length, weighs 0.1kg, and its circular plane of light reflection is 12cm in diameter; this has been located at a distance of -2cm in the direction of the pendulum bar; the material used was of solid and light characteristic (for the case was cardboard paper), so as not to significantly affect the dynamics of the pendulum. Since the pendulum will have an equivalent oscillation period of T = 2π √ (l/g) equivalent to 1.09 seconds, a sampling rate of 20 samples per second is adopted. Additionally, it has been considered essential to include a filter in the signal acquired from the sensor to minimize noise. The filter used corresponds to the FIR filter coefficient of order 5, which yielded a very acceptable result in terms of noise cancellation for the signal from the TCRT5000 sensor. The code of the developed program is shown in Appendix A.”
In addition, on page 17, lines 345 to 352, the following explanation was included:
“To fulfill this requirement, a mechanism based on a spring that has been adjusted to print a tangential linear velocity at the instant that includes a theta angle of 45 degrees was used (Figure 14). The required linear velocity corresponds to the value resulting from the product of ω and its perpendicular distance to the vertical axis of rotation, which is equivalent to a radius of r = l ∗ sin(45). This implies a velocity of 2(0.3sin(45)), corresponding to approximately 0.42m/s. It should be emphasized that this experiment aims to capture, using the sensors, an approximate movement similar to the simulations in case two and, in this way, to record the sensor readings for later analysis.”
- In Figure 9, it is necessary to locate the frames, as well as the locations of Sx and Sy, which appear different from the case shown in Figures 7 and 8.
Figures 9 and 10 were corrected so that they contain the location and label information for each of the sensors.
- It is mentioned in the conclusions that to quantify the damping factor β practically, prolonged experimentation was performed. It is perceived throughout the reading that no repeated experiments were carried out. It is necessary to clarify this point. In case experiments have not been repeated, they should be repeated, and the relevant analysis of results should be carried out. The same applies to all experiments.
It can be emphasized that the adoption of factor B was through the observation of the signal decay trend while the experiment was being developed and considering that its incorporation in the mathematical model allowed the development of the simulations by contemplating a mathematical model that is not ideal but closer to reality considering that the system is affected by the friction of the medium.
The following paragraph is included in the conclusions section on page 22, lines 459 to 462
“It is essential to note that the adoption of factor β will enable the observation of the damping tendency of the medium in which the movement develops. This aspect is not of primary interest. However, it is necessary to include it in the mathematical model for simulation purposes.”
In another instance, the primary focus of our research is emphasized, which involves the calculation of the pendulum angles based on the distances measured by the sensor and the approach of inverse kinematics using Denavit Hartemberg through homogeneous matrices. The text was included on page 21, lines 434 to 436, as follows:
“using the Denavit-Hartenberg method, who avoids making geometric deductions and instead adopts a matrix and vector technique based on the concept of homogeneous matrices.”
- From my point of view, observing similarity and movement trends is not enough to conclude that the method is valid. The measurements made must have a high degree of accuracy before this work could be published.
It is considered that the method can be validated only by statically positioning the pendulum system at different angles. This aspect was completed, but it was deemed scarce. Therefore, the idea of generating the dynamics by evaluating the trend was adopted, as the initial conditions, particularly the printing velocities, require further treatment in a new work focused on the control system. In this sense, a paragraph was included that summarizes what is described on page 22, lines 463 to 471, being:
“Finally, it is worth emphasizing that the exact difference between the experiments and the simulations that incorporate the theoretical calculation has not been quantified, as the generation of the initial conditions was not the primary purpose of this research. However, it is essential that the tendency of movement for each case remains dependent and allows for validation of the proposed sensing method. Although the effectiveness of the method can be verified by statically positioning the pendulum at different points, it is a very elementary approach; therefore, its validation in a moving environment and with the dynamics of the mechanism itself is considered enriching. In this sense, a new window of research is opened involving the performance of the pendulum system and its motion control. “
Kind regards
Marco Carpio

Reviewer 3 Report
Comments and Suggestions for Authors
Thie manuscript address an interesting topic, from a theoretical and an experimental point of view. It is well written and easy to follow.
I suggest a few improvements, in order to better explain the presented results.
- In the introduction, authors can report explicitly some practical examples where the proposed non-invasive technique can be useful.
- The MATLAB code used in section 3 is made available? That woulb in agreement with the open science model, and it would improve the interest of the readers.
- Remove the spanish text in section 4
- Reorganize the bullet lists in section 4.1 and 4.2
- Discuss further how the resolution of the TCRT5000 is limiting the setup operation
Author Response
Dear Reviewer, we, the authors, commend your contribution through the recommendations issued by you, which undoubtedly enrich our research and motivate us to continue.
Each observation has been analyzed, and we have worked to give it the best attention. In addition, the quality of the images has been improved, and the translation has been revised. We report in detail the adjustments made to the document as follows:
- In the introduction, authors can report explicitly some practical examples where the proposed non-invasive technique can be useful.
The introduction has been strengthened from page 1, line 25 to page 2, line 34, emphasizing the applications and differences between traditional measurement methods (such as potentiometers and encoders) and the infrared sensor solution in terms of accuracy, invasiveness, and long-term stability, the text included required the addition of five bibliographic sources, the text added is:
“Specifically, methods incorporating potentiometers and encoders have been widely used due to their simplicity and high accuracy in control and monitoring applications [7], [8]. These devices present significant limitations in terms of invasiveness and long-term stability. In particular, potentiometers require direct physical contact, which makes them invasive solutions, especially problematic in biomedical or high-sensitivity applications, where contact may alter the system or compromise its integrity [9], [10]. Additionally, both potentiometers and encoders are susceptible to mechanical wear, contaminant accumulation, and temperature fluctuations, which can impact accuracy and necessitate periodic maintenance or calibration to maintain optimal performance [8].”
In addition, on page 2, between lines 41 to 47, another text justifying the method was included, as follows.
“These sensors, especially those based on infrared spectroscopy or proximity beams, have demonstrated high accuracy comparable to that of traditional methods, with the advantage of eliminating the physical wear and tear inherent to contact systems. However, their long-term stability can be affected by instrumental drift and environmental conditions, so the implementation of online calibration and dynamic compensation techniques, such as orthogonal projection, becomes essential to maintain their reliability in special applications [10], [16].”
The titles of the references included are:
- A conceptual development of novel ultra-precision dimensional measurement technology
- In Situ Drift Monitoring and Calibration of Field-Deployed Potentiometric Sensors Using Temperature Supervision
- On-line recalibration of spectral measurements using metabolite injections and dynamic orthogonal projection
- Invasive and non-invasive measurement in medicine and biology: Calibration issues
- The Good pH probe: non-invasive pH in-line monitoring using Good buffers and Raman spectroscopy
- ¿ The MATLAB code used in section 3 is made available? That woulb in agreement with the open science model, and it would improve the interest of the readers.
A text is inserted detailing certain important variables considered in the code, including the sampling frequency chosen according to the natural period of the pendulum and the characteristics of the filter used for the treatment of the signal acquired from the sensor, in addition to the appendix A containing the MATLAB code developed and that could be useful at the moment of recreating the experimentation.
The text included on page 15, line 295 to 304 is:
“The prototype measures 0.3m in length, weighs 0.1kg, and its circular plane of light reflection is 12cm in diameter; this has been located at a distance of -2cm in the direction of the pendulum bar; the material used was of solid and light characteristic (for the case was cardboard paper), so as not to significantly affect the dynamics of the pendulum. Since the pendulum will have an equivalent oscillation period of T = 2π √ (l/g) equivalent to 1.09 seconds, a sampling rate of 20 samples per second is adopted. Additionally, it has been considered essential to include a filter in the signal acquired from the sensor to minimize noise. The filter used corresponds to the FIR filter coefficient of order 5, which yielded a very acceptable result in terms of noise cancellation for the signal from the TCRT5000 sensor. The code of the developed program is shown in Appendix A.”
- Remove the spanish text in section 4 .
The text that was mistakenly left without translation has been corrected. Page 9, lines 172 to 174
- Reorganize the bullet lists in section 4.1 and 4.2.
It was corrected, and a redundant item was eliminated.
- Discuss further how the resolution of the TCRT5000 is limiting the setup operation.
A paragraph is included in the conclusions detailing characteristics of the sensor, such as the distance that can guarantee a measurement, being a fundamental aspect to take into account when positioning the sensors and the plane of reflection. The text is inserted on page 21, lines 424 to 431.
“The sensors are used to measure distance, providing an analog signal proportional to the distance. This signal must be digitally processed using noise filters to achieve a more reliable measurement. Despite this, the sensors proved helpful in this investigation. Being a main component, these could be improved with better performance sensors for industrial applications, considering that the TCRT5000 sensor has a limitation in terms of the detection reflection distance that goes from 1mm to 25mm, this implies that the reflection plane must be positioned at a distance less than 25mm, therefore those oscillations that exceed this distance will present inconsistencies.”
Kind regards
Marco Carpio

Round 2
Reviewer 2 Report
Comments and Suggestions for Authors
Dear authors,
Thank you for addressing my comments. I have no further comments.
Reviewer 3 Report
Comments and Suggestions for Authors
Authors have addressed all my remarks.